# Role of Vacha (*Acorus calamus* Linn.) in Neurological and Metabolic Disorders: Evidence from Ethnopharmacology, Phytochemistry, Pharmacology and Clinical Study

**DOI:** 10.3390/jcm9041176

**Published:** 2020-04-19

**Authors:** Vineet Sharma, Rohit Sharma, DevNath Singh Gautam, Kamil Kuca, Eugenie Nepovimova, Natália Martins

**Affiliations:** 1Department of Rasa Shastra and Bhaishajya Kalpana, Faculty of Ayurveda, Institute of Medical Sciences, BHU, Varanasi, Uttar Pradesh 221005, India; vinitbhu93@gmail.com (V.S.); drdnsgautam@gmail.com (D.S.G.); 2Department of Chemistry, Faculty of Science, University of Hradec Králové, Rokitanskeho 62, 50003 Hradec Králové, Czech Republic; eugenie.nepovimova@uhk.cz; 3Faculty of Medicine, University of Porto, Alameda Prof. Hernani Monteiro, 4200-319 Porto, Portugal; 4Institute for research and Innovation in Heath (i3S), University of Porto, Rua Alfredo Allen, 4200-135 Porto, Portugal

**Keywords:** *Acorus calamus*, ethnomedicinal, phytochemistry, toxicity, pharmacological action, clinical trial, neuroprotective, neurological, metabolic application

## Abstract

Vacha (*Acorus calamus* Linn. (Acoraceae)) is a traditional Indian medicinal herb, which is practiced to treat a wide range of health ailments, including neurological, gastrointestinal, respiratory, metabolic, kidney, and liver disorders. The purpose of this paper is to provide a comprehensive up-to-date report on its ethnomedicinal use, phytochemistry, and pharmacotherapeutic potential, while identifying potential areas for further research. To date, 145 constituents have been isolated from this herb and identified, including phenylpropanoids, sesquiterpenoids, and monoterpenes. Compelling evidence is suggestive of the biopotential of its various extracts and active constituents in several metabolic and neurological disorders, such as anticonvulsant, antidepressant, antihypertensive, anti-inflammatory, immunomodulatory, neuroprotective, cardioprotective, and anti-obesity effects. The present extensive literature survey is expected to provide insights into the involvement of several signaling pathways and oxidative mechanisms that can mitigate oxidative stress, and other indirect mechanisms modulated by active biomolecules of *A. calamus* to improve neurological and metabolic disorders.

## 1. Introduction

Globally, an estimated 450 million people are suffering from mental disorders and about 425 million are known diabetics [1,2]. In 2016, 650 million adults were obese and about 23.6 million people were estimated to die of cardiovascular diseases (CVDs) by the year 2030 [3]. Metabolic disorders are characterized by hypertension, hyperglycemia, abdominal obesity, and hyperlipidemia, which may worsen the neurological disease risk. Improper diet (high calorie intake), lifestyle (e.g., smoking, chronic alcohol consumption, sedentary habits), and/or low level of nitrosamines (through processed food, tobacco smoke, and nitrate-containing fertilizers) affect the liver and can further lead to fatty liver disease [4,5]. In this condition, fatty changes may be due to increased production or decreased use of fatty acids, which may lead to inflammatory injury of hepatocytes, where inflammatory mediators, such as cytokines and interleukins, are released, which, along with lower adipokines, may eventually develop hepatic insulin resistance [6]. The same pathology also mediates diabetes, obesity, and peripheral insulin resistance. Insulin resistance also promotes the release of ceramides and other toxic lipids which enter the circulation and cross the blood–brain barrier leading to brain insulin resistance, inflammatory changes, and further progression to neurodegeneration and neurological disorders (Figure 1) [7].

*Acorus calamus* Linn. (Acoraceae), also known as Vacha in Sanskrit, is a mid-term, perennial, fragrant herb which is practiced in the Ayurvedic (Indian traditional) and the Chinese system of medicine. The plant’s rhizomes are brown in color, twisted, cylindrical, curved, and shortly nodded. The leaves are radiant green, with a sword-like structure, which is thicker in the middle and has curvy margins (Figure 2) [8]. Several reports ascertained a wide range of biological activities involving its myriad of active phytoconstituents. In this sense, the intent of this review is to assemble and summarize the geographical distribution, ethnopharmacology, phytochemistry, mechanism of action of *A. calamus* along with preclinical and clinical claims that are relevant to manage neurological and metabolic disorders. To the best of our knowledge, so far, none of the published reviews has described all the characteristics of this medicinal plant [9,10,11]. The present report is expected to produce a better understanding of the characteristics, bioactivities, and mechanistic aspects of this plant and to provide new leads for future research.

## 2. Methodology

The literature available in the Ayurvedic classical texts, technical reports, online scientific records such as SciFinder, Google Scholar, MEDLINE, EMBASE, Scopus directory were explored for ethnomedicinal uses, geographical distribution, phytochemistry, pharmacology, and biomedicine by applying the following keywords: “*Acorus calamus*”, “Vacha”, “Medhya”, “neuroprotective”, “phytochemistry”, “obesity”, “oxidative stress”, “anticonvulsant”, “antidepressant”, “antihypertensive”, “anti-inflammatory”, “immunomodulator”, “antioxidant”, “diabetes”, “mechanism of action” with their corresponding medical subject headings (MeSH) terms using conjunctions OR/AND. The search was focused on identifying Ayurvedic claims in the available ethnomedicinal, phytochemical, preclinical, clinical, and toxicity reports to understand the role of *A. calamus* in neurological and metabolic disorders. This search was undertaken between January 2018 and January 2020. Searches were restricted to the English language. The search methodology as per the Preferred Reporting Items for Systematic Reviews and Meta-Analysis (PRISMA) is stipulated in the flowchart in Figure 3.

## 3. Geographical Distribution

*A. calamus* grows in high (1800 m) and low (900 m) altitudes and it is found to be geographically available in 42 countries [8]. Furthermore, as per the Global Biodiversity Information Facility records [12], the distribution of this plant in several parts of the world, as well as in India, is highlighted in Figure 4.

## 4. Ethnomedicinal Use

This plant is being practiced traditionally in the Indian Ayurvedic tradition, as well as in the Chinese system of medicine for analgesic, antipyretic, tonic, anti-obesity, and healing purposes; it is highly effective for skin diseases, along with neurological, gastrointestinal, respiratory, and several other health disorders. Rhizomes and leaves are found to be profusely practiced in the form of infusion, powder, paste, or decoction [13,14,15,16,17,18,19,20,21,22,23,24,25,26,27,28,29,30,31,32,33,34,35,36,37,38,39,40,41,42,43,44,45,46,47,48,49,50,51,52,53,54,55,56,57,58,59,60,61,62,63,64,65,66,67,68,69,70,71,72]. The ethnomedicinal uses of the *A. calamus* are detailed in Table 1.

*A. calamus* rhizomes and leaves are also used as an active pharmaceutical ingredient in various Ayurvedic formulations (Table 2).

## 5. Phytochemistry

The phytochemical investigation of this plant has been ongoing since the year 1957 [73,74]. To date, about 145 compounds were isolated from *A. calamus* rhizomes and leaves, viz. phenylpropanoids, sterols, triterpene glycosides, triterpenoid saponins, sesquiterpenoids, monoterpenes, and alkaloids (Table 3). Amongst those, phenylpropanoids (chiefly, asarone and eugenol) and sesquiterpenoids have been considered the principal effective compounds of *A. calamus*. Chemical structures of isolated compounds from *A. calamus* are illustrated in Figure 5.

### 5.1. Phenylpropanoids

Phenylpropanoids have an aromatic ring with a structurally diverse group of phenylalanine-derived secondary plant metabolites (C_6_–C_3_), like *α*-asarone, *β*-asarone, eugenol, isoeugenol, etc. [75]. A number of phenylpropanoids have been identified from *A. calamus* rhizome and leaves **(1-45)**. *α* and *β*-asarone isolated from the rhizome are the predominant compounds present in this plant. A series of aromatic oils from the rhizome with diverse structures are also reported [74,75,76,77,78,79,80,81,82,83,84,85,86,87,88,89,90,91,92,93,94,95,96,97,98].

### 5.2. Sesquiterpenoids

About 44 sesquiterpenes, including lactones, were characterized and identified in *A. calamus* rhizomes. Sesquiterpene lactones are produced of 3 isoprene units and composed of lactone rings. *α*–*β* unsaturated *γ*-lactonic ring in sesquiterpene lactones is believed to be responsible for pharmacological activity (46-99) [74,78,89,91,93,98,99,100,101,102,103,104].

### 5.3. Monoterpenes

Monoterpenes (C-10) are the simplest class of the terpene series that belongs to two isoprene units (tricyclic, bicyclic, monocyclic, etc.). Monoterpenes can have different functional groups, like aldehydes, ketones, esters, ethers, phenols, and alcohols [80]. These organic compounds emit the characteristic flavor and fragrance of *A. calamus* leaves and rhizomes (100-122) [74,78,89,91,93,97,98].

### 5.4. Triterpenoid Saponins

Triterpenoid saponins are made up of a pentacyclic C-30 terpene skeleton as a pillar. Limited reports studying triterpenoid saponins in *A. calamus* are available, and only two triterpenoid saponins (124, 125) have been isolated from *A. calamus* rhizomes (Table 3) [85].

### 5.5. Other Compounds

To date, one xanthone glycoside (123) [82,83], two alkaloids (126-127) [84], one triterpene glycoside (128), one steroid (129) [85], 12 amino acids (130-141) [86,87], and 4 fatty acids (142-145) [88] have been identified in *A. calamus* rhizomes [83,84,85,86,87,88].

## 6. Pharmacological Properties

Diverse bioactivities of *A. calamus* extracts are evident from preclinical (in vitro and in vivo) and clinical reports, such as antidiabetic, anti-obesity, antihypertensive, antioxidant, anti-inflammatory, immunomodulatory, anticonvulsant, and neuroprotective [105,106,107,108,109,110,111,112,113,114,115,116,117,118,119,120,121,122,123,124,125,126,127,128,129,130,131,132,133,134,135,136,137,138,139,140,141,142,143,144,145,146,147,148,149,150,151,152,153,154,155,156,157,158,159,160,161,162,163,164,165,166,167,168,169,170,171,172,173]. The summarized information on *A. calamus* botanical parts, extract type, and their bioactivities in neurological and metabolic disorders is stipulated in Table 4.

### 6.1. Antidiabetic Effect

The antidiabetic effect of *A. calamus* ethyl acetate fraction was evaluated in streptozotocin (STZ)-induced and diabetic (db/db) mice. Glucagon-like peptide-1 (GLP-1) levels, plasma insulin, “and related gene expression were evaluated. The fraction (100 mg/kg, intragastric (i.g.)) indicated a significant reduction in blood glucose levels. For in vitro, at the concentration of 12.5 μg/mL, a significant increment in GLP-1 levels was found in the insulin-secreting L-cell culture medium [108]. The ethyl acetate radix fraction exhibited a significant effect on the HIT-T15 cell line and *α*-glucosidase enzyme. The ethyl acetate fraction also enhanced insulin secretion in HIT-T15 cells and blocked the *α*-glucosidase in vitro activity with 0.41 μg/mL of inhibitory concentration (IC_50_) [109].”

### 6.2. Anti-Obesity Effect

The *β*-asarone compound isolated from the rhizome was investigated against high-fat diet (HFD)-induced obesity in animals. *β*-Asarone-treated adipose rats showed weight loss, but also inhibited metabolic transformations, as well as glucose intolerance, elevated cholesterol, and adipokine variance [143]. The in vitro investigation on the *A. calamus* aqueous extract showed lipid-lowering activity through inhibition of the pancreatic lipase percentage (28.73%) [144].

### 6.3. Antihypertensive Effect

The antihypertensive effects of *A. calamus* were studied on their own, in isolation, and in combination with *Gymnema sylvestre* in the HFD-induced hypertension in rats. The HFD was given for 4 weeks, which significantly increased the average systolic blood pressure (SBP). At a 200 mg/kg dose, *A. calamus* in combination with *G. sylvestre* reduced the SBP and heart rate significantly. *A. calamus* with *G. sylvestre* exhibited synergistic effect as compared with individual herbs [145].

### 6.4. Anti-Inflammatory and Immunomodulatory Effect

The methanolic *A. calamus* rhizome extract (12.5 µg/mL) prevented the VCAP-1 and intercellular expression on the surface of mouse myeloid leukemia cells and murine endothelial cells, respectively [146]. In an in vitro anti-inflammatory study (Red blood cell membrane stabilization method), the *A. calamus* aqueous rhizome extract at the highest concentration of 10 mg/mL showed insignificant activity against hemolysis inhibition and the RBC membrane stabilization percentage [144]. Aqueous *A. calamus* leave extract was studied on HaCaT cells and restricted the characteristics of interleukin (IL)-8, IL-6 RNA protein levels alongside interferon regulatory factor 3 (IRF3) and nuclear factor kB (NF-κB) activation [147]. N-hexane, butanolic, and aqueous fractions of *A. calamus* were evaluated against cyclooxygenase (COX) and lipoxygenase (LOX)-mediated eicosanoid production by arachidonic acid. The butanolic fraction inhibited the COX-mediated production of thromboxane B2 (TXB2) and lipoxygenase product 1 (LP1). Investigation of the underlying signaling pathways revealed that the butanolic fraction inhibited phospholipase C (PLC) pathway in platelets, presumably acting on protein kinase C (PKC) [148]. The essential oil isolated from *A. calamus* was evaluated by protein denaturation assay, where at the concentration level of 300 μg/mL, 69.56% of the inhibition level was observed [149].

### 6.5. Antioxidant Effect

The in vitro antioxidant activity of acetone, acetonitrile, alcoholic, and aqueous extracts of *A. calamus* rhizomes exhibited free radical scavenging activity on the [2,2′-azinobis (3-ethylbenzothiazoline-6-sulphonic acid)] free radical scavenging activity assay (ABTS), the (1, 1-diphenyl-2-picrylhydrazyl) free radical scavenging activity assay (DPPH), and the ferric ion reducing antioxidant power assay (FRAP). Strong antioxidant effect was noticed in the acetone extract, followed by acetonitrile and methanol, while in the aqueous extract, poor antioxidant activity was found [150]. The aqueous extract exhibited superior antioxidant effects in metal ion chelation, lipid peroxidation (LPO), and DPPH assays [144,151]. The in vitro antioxidant activity of ethanol, hydro-ethanol, and aqueous whole plant extracts of *A. calamus* was investigated using FRAP, DPPH, nitric oxide, hydroxyl radical, reductive ability, and superoxide radical scavenging activity. The existence of phenolics and flavonoids in *A. calamus* are believed to contribute to the promising antioxidant effect. IC_50_ values of the ethanol extract were found to be 54.82, 109.85, 38.3, 118.802 µg/mL for the scavenging activities of DPPH, hydroxyl radical, superoxide radical, and nitric oxide, respectively. The irreversible potential of the above results and the FRAP values of the extracts were found to augment in a concentration-dependent manner [152]. “Ethanol and hydro-alcoholic extracts of *A. calamus* roots and rhizomes were studied for antioxidant potential against DPPH compared with butylated hydroxyanisole (BHA) and silymarin. Ethanol and hydro-alcoholic extracts showed free radical scavenging activity of 59.13 ± 18.95 and 56.71 ± 19.54, respectively [153,154,155]. The essential oil isolated from *A. calamus* showed strong antioxidant efficacy against the *β*-carotene/linoleic acid bleaching test and DPPH free radicals [156]. The methanol extract of the *A. calamus* rhizome was evaluated against the free radical scavenging activity, and the reported IC_50_ value was 704 µg/mL [157]. The IC_50_ of the essential oil was 1.68 μg/mL, which showed virtuous free radical scavenging activity in the DPPH test [149].”

### 6.6. Anticonvulsant Effect

The methanol extract shows anticonvulsant effects feasibly through potentiating the action of gamma-aminobutyric acid (GABA) pathway in the central nervous system [124]. When it comes to the purification of *A. calamus* rhizome in cow urine, it is advocated in the Ayurvedic pharmacopoeia of India (API) before its therapeutic use. The purified rhizome was investigated in a maximal electroshock (MES) seizure model, and phenytoin was used as the standard drug. The raw and processed rhizome (11 mg/kg, p.o.) exhibited notable anticonvulsant activity by minimizing the span of the tonic extensor period in rats, whereas the processed rhizome showed better therapeutic activity than when it was raw [158]. The calamus oil isolated from the *A. calamus* rhizome was evaluated at varying dose levels of 30, 100, and 300 mg/kg, p.o., body weight (b.w.), against MES, pentylenetetrazol (PTZ), and minimal clonic seizure (MCS) models. The calamus oil was found to be neurotoxic at 300 mg/kg, though it was effective in the MCS test at 6 Hz. The protective index value of calamus oil was found to be 4.65 [125].

### 6.7. Antidepressant Effect

Interaction of the methanolic *A. calamus* rhizome extract with the adrenergic, dopaminergic, serotonergic, and GABAergic system was found responsible for the expression of antidepressant activity [128]. In another study, the methanolic *A. calamus* leave extract showed significant activity through a reduction in the immobility period in the TST and FST [129]. Through interaction with the adrenergic and dopaminergic system, the hydro-alcoholic extract was normalized to the over-activity of the hypothalamic pituitary adrenal (HPA) axis [131]. Sobers capsules (a herbo-mineral formulation containing *A. calamus*) were evaluated by tail suspension and forced swimming tests in mice. At the oral dose of 50 mg/kg for 14 days, capsules exhibited insignificant impact on locomotor activity, and caused antidepressant effects in experimental animals [159]. Tensarin (the traditional medicine of Nepal containing *A. calamus*) was evaluated for the anxiolytic effect in mice using the open field test (OFT), activity monitoring along with the passive avoidance test. At all three dose levels (50, 100, 200 mg/kg), Tensarin produced an anxiolytic effect in a dose-dependent way by an improvement in rearing, number of passages, and duration of the period employed by mice [160].

### 6.8. Neuroprotective Effect

The ethanolic extract was studied (25, 50, and 100 mg/kg doses, oral and intraperitoneal routes) for learning and memory-enhancing activity. The subjects used consisted of male rates, through Y maze and shuttle box tests models. The findings showed an increase in acquisition–recalling and spatial recognition data [161]. The ethanolic *A. calamus* rhizome extract (0.5 mL/kg, i.p.) potentiated pentobarbitone-created sleep periods, which caused significant inhibition of conditioned avoidance response in rats and marked (40–60%) protection against PTZ-induced convulsions, although it did not show any spontaneous motor activity and impact the aggressive or fighting behavior response in male rat pairs [162].

### 6.9. Cardioprotective Effect

The alcoholic *A. calamus* rhizome extract (100 and 200 mg/kg) considerably attenuated isoproterenol-led cardiomyopathy in rats and showed a significant reduction in the heart/body weight ratio, level of serum calcineurin, serum nitric oxide, serum lactate dehydrogenase (LDH), and thiobarbituric acid reactive substances (TBARS) level. However, the level of the antioxidant enzyme was found increased at the 100 mg/kg extract dose level [163]. The crude extract and its fractions (0.01–10 mg/mL) were investigated in an isolated rabbit heart, which showed mild reduction in the force of forced vital capacity (FVC), hazard ratio (HR), and cystic fibrosis (CF), while the ethyl acetate extract exhibited complete suppression, and the n-hexane fraction showed the same effect on FVC and HR, but enhanced CF. The extract and its fractions exhibited controlled coronary vasodilator effect, interceded maybe by an endothelial-derived hyperpolarizing factor [164]. The cardioprotective potential of the whole plant’s ethanolic extract (100 and 200 mg/kg) reduced serum enzyme levels and shielded the myocardium from the lethal effect of DOX [141].

### 6.10. Cytochrome Inhibitory Activities

Cytochromes P450 (CYPs) are the prime enzymes that catalyze the oxidative metabolism of a wide variety of xenobiotics. It is known that 2,4,5-trimethoxycinnamic acid is the main metabolite of α- or β- asarone [165]. The metabolism rate of α- and β-asarone was shown to be directly proportional to the CYPs concentration in rat hepatocytes and liver microsomes [166,167]. CYP3A4 (CYP isoforms) has been reported for bioactivation of α-asarone [168]. The hydro-alcoholic *A. calamus* extract and α-asarone were evaluated by the CYPs-carbon monoxide complex method. The extract exhibited moderate potential interaction in CYP3A4 (IC_50_ = 46.84 μg/mL) and CYP2D6 (IC_50_ = 36.81 μg/mL), while α-asarone showed higher interaction in CYP3A4 (IC_50_ = 65.16 μg/mL) and CYP2D6 (IC_50_ = 55.17 μg/mL) [169]. These outcomes indicated that both extracts and α-asarone interacted quite well in drug metabolism and also had an inhibitory effect on CYP3A4 and CYP2D6. The drug-drug interaction effect of the *A. calamus* extract and its main chemical constituent (α and β-asarone) needs to be studied in more CYPs isomers, like CYP2C9 and CYP2E1.

### 6.11. Toxicity and Safety Concerns

In acute and sub-acute toxicity of the hydro-alcoholic extract of *A. calamus* in rats, at the highest dose level of 10 gm/kg, no severe changes were observed, and the lethal dose (LD_50_) was found to be 5 g/kg [170]. The petroleum ether extracts (obtained by cold rolling, water distillation, and Soxhlet extraction methods) of the *A. calamus* rhizome showed mild toxicity in two-day-old oriental fruit flies [171]. The ethanolic extract of the *A. calamus* rhizome at oral dosage of 175, 550, 1750, and 5000 mg/kg b.w. was given for 14 days within an acute toxicity study, while at the dose level of 0, 200, 400, and 600 mg/kg, p.o., the extract was given for 90 days within a chronic toxicity study. At the doses of 1750 and 5000 mg/kg, piloerection, tremors, and abdominal breathing were found for 30 min [172]. In that study, *A. calamus* was purified for 3 h in cow urine, decoction of *Sphaeranthus indicus*, and decoction of leaves of *Mangifera indica*, *Eugenia jambolana*, *Feronia limonia*, *Citrus medica*, and *Aegle marmelos*, followed by fomentation with Gandhodaka (decoction of six aromatic herbs) for 1 h. The acute oral toxicity test of raw and purified *A. calamus* was performed in albino rats at 2000 mg/kg for 2 weeks. At the 2000 mg/kg dose, *A. calamus* did not produce any toxic symptoms within 14 days [173].

The *β*-asarone compound isolated from *A. calamus* was found to be carcinogenic and toxic [174]. The LD_50_ value of *β*-asarone by oral and intraperitoneal route was found to be 1010 and 184 mg/kg, respectively, in mice and rats [175]. The LD_50_ of calamus oil was found to be 8.88 gm/kg b.w. [176], while in the calamus oil obtained from Jammu, India, the LD_50_ was 777 mg/kg b.w. [177]. Overall, several investigations have been carried out on *A. calamus* regarding its toxicity; however, no noticeable data on toxicity have been found so far.

## 7. Clinical Reports

*A. calamus* has also been clinically investigated as a monotherapy as well as in combination with other medicinal herbs in healthy subjects and sufferers of various metabolic and neurological ailments. Most clinical research has looked at the *A. calamus* effect on obesity, depression, neuroprotection, and cardiovascular disease [178,179,180,181,182,183,184,185,186,187,188,189,190,191]. The data obtained so far can be found in Table 5. Furthermore, a systematic review reveals that *A. calamus* (alone or in combination therapy) exhibits anti-obesity, antidepressant, and cardioprotective effects, as well as helps physical and mental performance.

## 8. Mechanistic Role

The proposed mechanism of action of *A. calamus* in neurological and metabolic disorders includes a synergic integration of antioxidant defense, GABAergic transmission, brain stress hormones modulation, pro-inflammatory cytokines, leptin and resistin levels, adipocytes inhibition, calcium channel blocker effect, protein synthesis, oxidative stress, acetylcholinesterase (AChE) inhibition, and anti-dopaminergic properties. A compendium of mechanisms of action of *A. calamus* in neurological and metabolic protection is illustrated in Figure 6 and Table 6. *A. calamus* significantly affects fasting blood sugar, insulin resistance, HbA1c, and the adipogenic transcription expression factor through various mechanisms, viz. antioxidant, anti-inflammatory, β-cells regeneration, improving insulin sensitivity, gluconeogenesis, nicotinamide adenine dinucleotide phosphate (NADPH) oxidase, and glucose transporter type 4 (GLUT-4)-mediated transport inhibition.

The antihypertensive effect of *A. calamus* may be explained by Ca^2+^ antagonists that affect the nitric oxide pathway. The chemical constituents of *A. calamus* upregulate the antioxidant effect, suppress pro-inflammatory cytokines, and act as detoxifying enzymes through the NF-κB and nuclear factor erythroid 2-related factor 2 (Nrf2) signaling pathways. The Nrf2 pathway may be activated by phenylpropanoids, sesquiterpenoids, and monoterpenes by interaction of active phytoconstituents with nitric oxide derivatives react with thiol groups between KEAP1 and Nrf2, along with Nrf2 phosphorylation. “When Nrf2 is released from the Kelch-like erythroid-derived CNC (cap’n’collar) homology protein (ECH)-associated protein 1 (KEAP1), it transfers into the nucleus, where it induces the genes encoding protein expression impenetrable in glutathione (GSH) synthesis, antioxidant, and detoxifying phase 2 enzymes. Oxidative stress and ligands for tumor necrosis factor receptors (TNFRs) and toll-like receptors (TLRs) activate upstream Ik-B kinases (IKKs), ensuing phosphorylation of IkB that is generally bound to the inactive NF-kB dimer in the cytoplasm. After that, IkB is targeted for proteasomal degradation and NF-kB, then it moves into the nucleus where it induces inflammatory cytokine expression in addition to the genes encoding proteins like superoxide dismutase (SOD) 2 and B cell chronic lymphocytic leukemia (CLL)/lymphoma 2 (Bcl2) involved in adaptive stress response (Figure 7). The bioactive molecules of *A. calamus* can inhibit NF-kB in inflammatory immune cells, while other phytoconstituents may activate NF-kB in neuronal cells to improve stress resistance.” *A. calamus* phytoconstituents regulate NF-kB, LOX, and COX-2 activity. These compounds dose-dependently suppress the production of inflammatory factors like NO, TNF-α, IL-6, IL-1β, and JNK signaling, acting as anti-inflammatory agents. In addition, it was also noted that the inflammation induced by various chemicals was inhibited by bioactive constituents through suppression of IkB/NF-kB and JNK/AP-1 signaling pathways. Thus, over several studies, it has been reported that asarone compounds have a potential against neurodegenerative diseases.

PPAR gene and C/EBP are involved in the differentiation process. PPAR-δ and PPAR-γ promote adipogenesis. In the same way, amino acids and glucose react with C/EBP- δ and C/EBP-β. If low levels of glucose induce gadd153, the inactive dimer is formed, with C/EBP-β inhibiting the progress of adipocyte development. C/EBP delta activates C/EBP-α. This is mainly involved in the formation of mature adipocytes and lipid accumulation in adipose tissue. In 3T3-L1 preadipocytes, *α*-asarone and *β*-asarone inhibited adipocyte differentiation and reduced the intracellular lipid accumulation, and also decreased the expression levels of adipogenic transcription factors (PPARγ and C/EBPα). These phytochemicals significantly promoted adenosine monophosphate-activated protein kinase (AMPK), which is known to suppress adipogenesis. It was also found that pretreatment with *α*-asarone and *β*-asarone, a typical inhibitor of AMPK, attenuated the inhibitory effect of asarone on AMPK phosphorylation. The asarone-induced AMPK activation leads to a decrease in adipogenic transcription factor expression, and suppresses adipogenesis.

## 9. Perspectives and Future Directions

The present review provides a plethora of information apropos ethnomedicinal uses, marketed formulations, geographical distribution, chemical constituents, pharmacological activities of crude, n-hexane, ethyl acetate, methanolic, ethanolic, hydro-alcoholic, aqueous extracts along with pure compounds, and clinical trials related to *A. calamus*.

Investigations on extracts and compounds of *A. calamus* suggested antidiabetic, anti-obesity, antihypertensive, anti-inflammatory, antioxidant, anticonvulsant, antidepressant, neuroprotective, and cardioprotective potentials with distinct underlying signaling pathways. The biological potential and mechanisms of action of some of the chemical constituents (*α*-asarone, *β*-asarone, eugenol) are known. However, other compounds need to be scientifically explored for their bioactivities and molecular modes of action, which could provide a lead for further development into therapeutics. More systematic, well-designed, and multi-center clinical studies are warranted to evaluate standardized extracts of *A. calamus* therapeutically and to identify the pharmacokinetic-dynamic roles of pharmacologically active biomolecules. There is scarce data from experimental and clinical reports on hypertension, diabetes, and atherosclerosis, and less supporting evidence is available on the use of *A. calamus* to treat hypertension and diabetes. Based on the available data, it is suggested that this plant could be used as an adjuvant to the established targeted drugs for neurological and metabolic disorders.

In 1974, United States food & drug administration (USFDA) banned *A. calamus* due to its carcinogenic effects following animal studies. They reported *β*-asarone as a carcinogenic agent, but the study was conducted on the calamus oil which consists of *β*-asarone in about 80%, while its different genotype in Europe and India contains *β*-asarone in lower concentrations. *A. calamus* cultivated in various geographical regions may have different chemical compositions along with therapeutic properties challenging quality control, toxicity, and safety concerns of *A. calamus*. In addition, the heavy metal, mycotoxin, and pesticide concentrations are required to be addressed in all toxicity studies.

## 10. Conclusions

Compelling in vitro, in vivo and clinical evidence suggests that the potential role of *A. calamus* rhizomes for modulating metabolic and neurological disorders could be due to their richness in several classes of active phytoconstituents. The predominant compounds present in rhizomes and leaves responsible for expression of potent bioactivities include *α*-asarone, *β*-asarone, eugenol, and calamine. The present report is expected to fill the gaps in the existing knowledge and could provide a lead for researchers working in the areas of phytomedicine, ethnopharmacology, and clinical research.

## Figures and Tables

**Figure 1 jcm-09-01176-f001:**
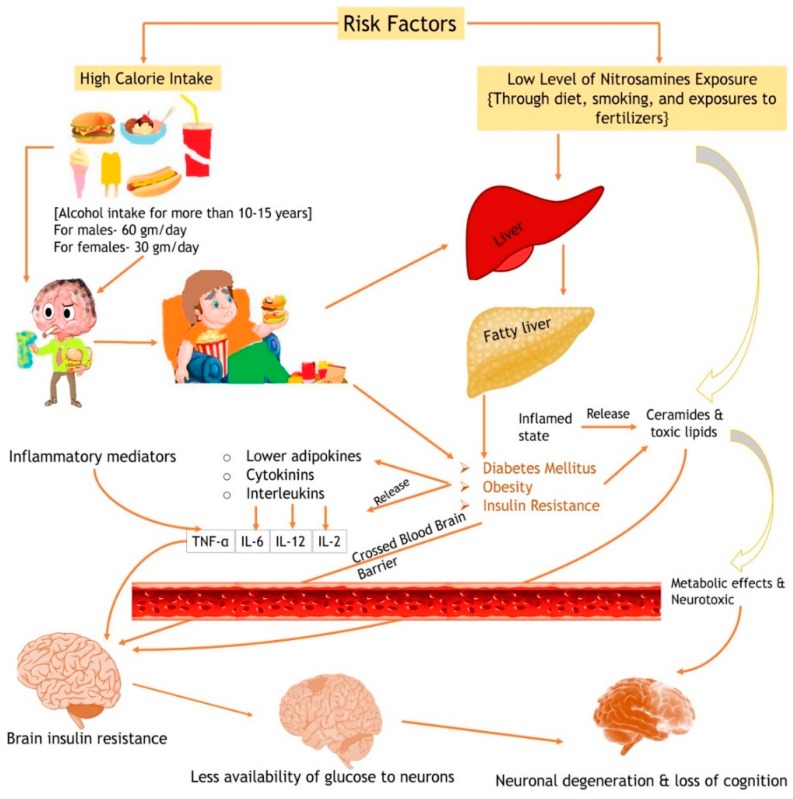
Pathophysiology of insulin resistance, metabolic malfunction, and progression to a neurological disorder. TNF, tumor necrosis factor; IL, interleukin.

**Figure 2 jcm-09-01176-f002:**
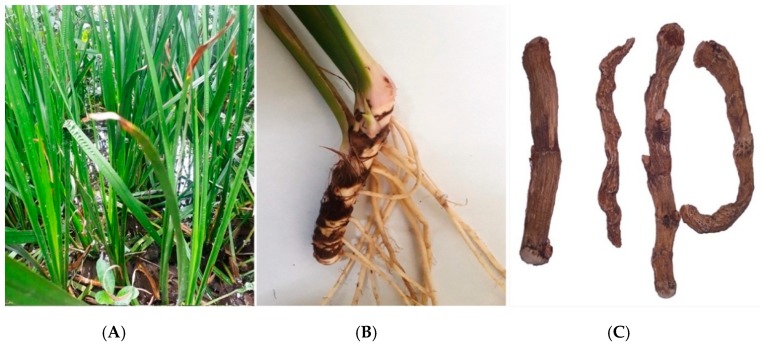
Photographs of *Acorus calamus*: (**A**) Natural habitat; (**B**) Fresh rhizome; (**C**) Dried rhizome.

**Figure 3 jcm-09-01176-f003:**
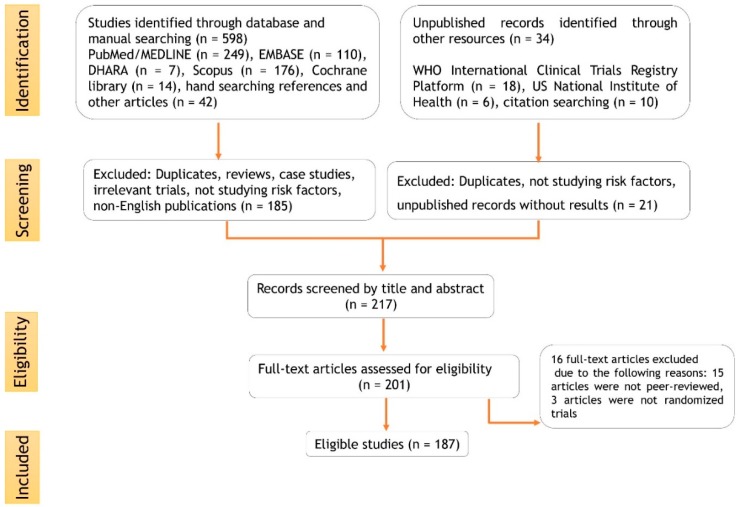
Flowchart of the selection process.

**Figure 4 jcm-09-01176-f004:**
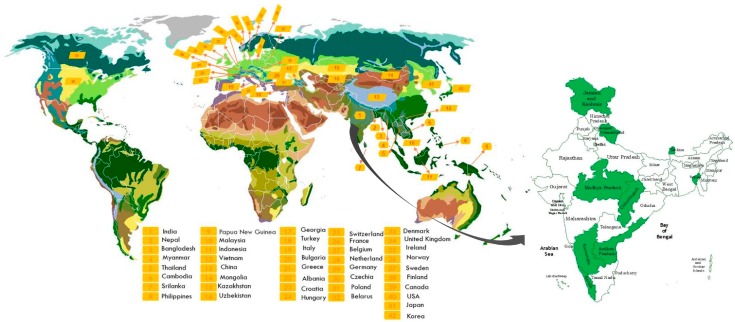
Distribution of *A. calamus* worldwide and in India.

**Figure 5 jcm-09-01176-f005:**
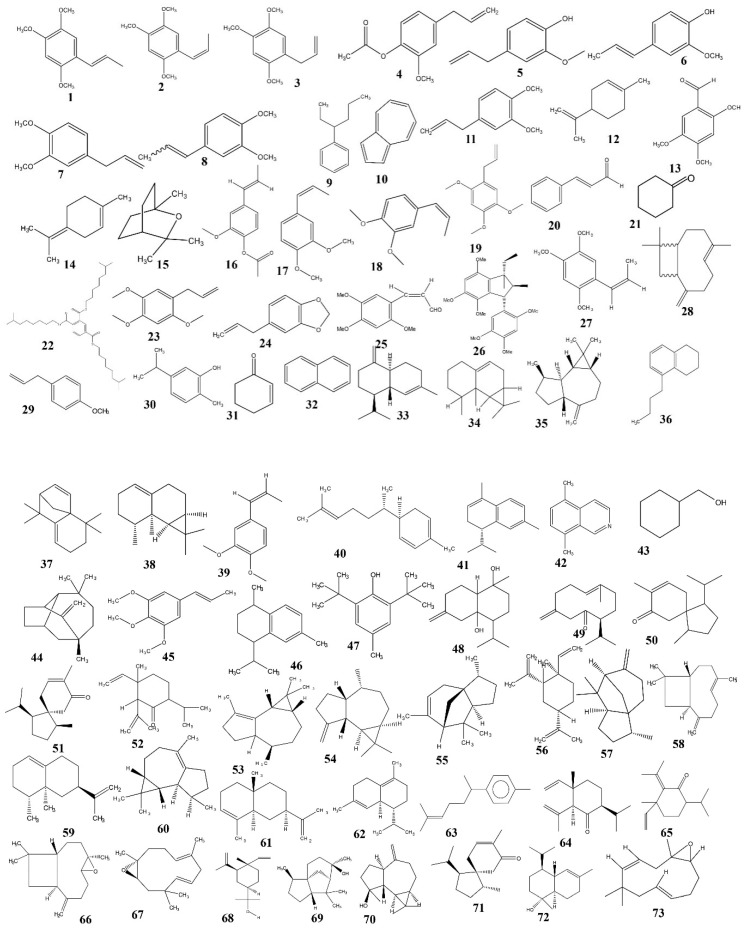
Chemical structures of isolated compounds from *A. calamus*.

**Figure 6 jcm-09-01176-f006:**
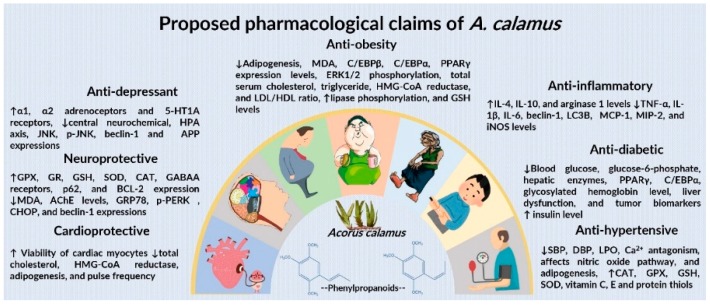
Illustration of role of *A. calamus* mechanisms in the treatment of neurological and metabolic disorders. AChE, acetylcholinesterase; APP, amyloid precursor protein; Bcl-2, B-cell lymphoma 2; CHOP, C/EBP homologous protein; CCAAT (cytosine-cytosine-adenosine-adenosine-thymidine)-enhancer-binding protein homologous protein; C/EBP, CCAAT enhancer-binding protein; GABAA, γ-Aminobutyric acid type A; GRP78, 78-kDa glucose-regulated protein; HMG-CoA, 3-hydroxy-3-methylglutaryl coenzyme A; iNOS, inducible nitric oxide synthase; JNK, c-Jun NH2-terminal kinase; LC3b, microtubule-associated proteins 1A/1B light chain 3B; MCP, modified citrus pectin; MDA, malondialdehyde; MIP, macrophage inflammatory protein; p-PERK, phospho-protein kinase RNA-like ER kinase; PPARγ, peroxisome proliferator-activated receptor gamma; ERK1/2, extracellular signal-regulated protein kinase.

**Figure 7 jcm-09-01176-f007:**
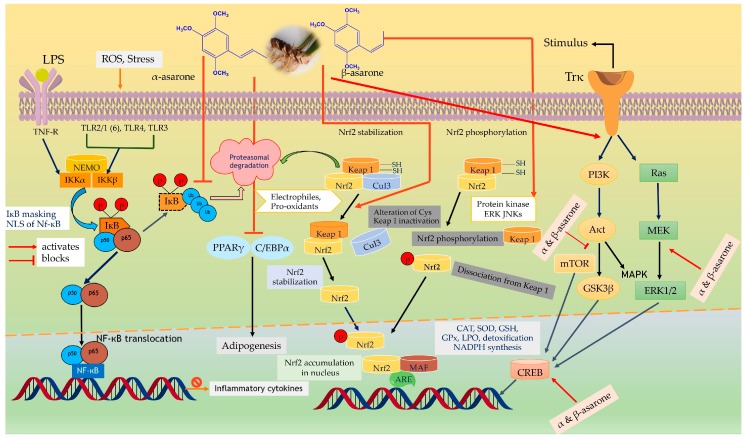
The role of the Nrf-2, NF-κB, PI3K/AKT, Ras/MAPK, and PPARγ signaling pathways as affected by phytoconstituents of *Acorus calamus* to upregulate antioxidant, neuroprotective, detoxifying enzymes and suppress inflammation. Ub, ubiquitin; NEMO, NF-kB essential modulator; ARE, antioxidant response element; Maf, musculoaponeurotic fibrosarcoma oncogene homolog; NLS, nuclear localization signal; CAT, catalase; GPX, glutathione peroxidase; Trk, tyrosine kinase receptor; LPS, lipopolysaccharide; TLRs, toll-like receptors; PI3K, phosphatidylinsoitol-3-kinase; MAPK, mitogen-activated protein kinase; mTOR, mammalian target of rapamycin; ERK, extracellular signal-regulated kinases; Nrf2, nuclear factor e2-related factor 2; Keap-1, kelch-like ECH-associated protein-1; MEK, mitogen-activated protein kinase; JNK, c-Jun N-terminal kinase;NADPH, nicotinamide adenine dinucleotide phosphate; NF-κB, nuclear factor-kappa B; IkB, inhibitor of kB; IKK, inhibitor of kB kinases.

**Table 1 jcm-09-01176-t001:** Ethnomedicinal use of *A. calamus* in various countries.

Country	Ailment/Use	Part Used/Dosage Form	Route of Administration	References
India	Eczema	The paste of *A. calamus* rhizomes are given with the paste of *Curcuma aromatica* rhizomes and *Azadirachta indica* leaves	Oral	[13]
	Skin diseases	Rhizomes paste *A. calamus* and *C. aromatica* are applied with the seed paste of *Argemone Mexicana*		
	Cough, stuttering, ulcer, fever, dermatitis, scab, sores	Rhizomes		[14]
	Cold, cough, and fever	Rhizomes paste of *A. calamus* is given to children with mother’s milk, *Myristica fragrance*, and *Calunarejan spinosa* fruits		[15]
		Two teaspoonfuls of herbal powder containing *A. calamus* rhizomes, *Boerhaavia diffusa* roots, *Calonyction muricutum* flower pedicles, *Ipomoea muricate* seeds, *Senna* leaves, *Cassia fistula* fruits pulp, *Curcuma longa* rhizomes, *Helicteres isora* fruits, and *Mentha arvensis* leaves, black pepper is taken with lukewarm water		[16]
	Gastric disorders	*A. calamus* rhizomes paste is given with cow milk		[17]
	Carminative, flavoring, tonic, and head lice infestation	Infusion of a dried rhizomes (collected and stored in the autumn season)		[17,18,19]
	Epilepsy, dysentery, mental illnesses, diarrhea, kidney and liver disorders	*A. calamus* rhizomes paste is given with honey		[20]
	Wounds, fever, body pain	Rhizomes		[21,22]
	Dysentery	Fresh ground rhizomes is mixed with hot water and given for 3 days		[23]
	Stimulant	Dry powder of *A. calamus* is given with honey		[24]
	Injuries	External application of the *A. calamus* rhizomes paste	Dermal	[25]
	Stomachache	Ash of the *A. calamus* rhizomes paste		[26]
	Otitis externa	*A. calamus* roots paste is given with coconut husk juice		[27]
	Lotion	Fresh leaves of *A. calamus*		[28]
	Cough, cancer, and fever	*A. calamus* roots juice is given with honey and *MyristicaDactyloides*	Oral	[29]
	Analgesic	*A. calamus* rhizomes are given with cinchona bark		[30]
	Gastrointestinal, respiratory, emmenagogue, antihelmintic	Rhizomes		
	Prolonged labor	Rhizomes is applied with saffron and horse milk		
	Paralysis, arthritis	Rhizomes ash is applied with castor oil		
	Neurological disorder, gastrointestinal, respiratory, increases menstrual flow, analgesic, contraceptive	Rhizomes	Oral	[31,32,33]
	Herpangina, analgesic, neurological disorder, gastrointestinal, respiratory			[34]
Pakistan	Colic and diarrhea	Whole plant		[35]
Nepal	Blood pressure	Roots infusion of *A. calamus*		[36]
	Cough, headache, snake bite, sore throat, and pain	Rhizomes		[37]
	Dysentery	Rhizomes juice is given with hot water		
	Neurological, respiratory	Rhizomes		[38]
Malaysia	Rheumatism, diarrhea, dyspepsia, and hair loss	Whole plant		[39]
Tibet	Fever, gastrointestinal	Dried rhizomes is given with *Saussurea lappa*, *Ferula foetida*, *Terminalia chebula*, *Cuminum cyminum*, *Inula racemosa*, and *Zingiber officinale*		[40]
	Cancer	Rhizomes		[41]
China	Gastrointestinal, respiratory, neuroprotective, analgesic, contraceptive, cancer	Rhizomes		[42,43,44]
	Antipyretic and ear-related disease	Rhizomes given with squeezed *Coccinia cordifolia* stems along with water		[45]
	Detoxification	Rhizomes with vinegar, *Alpinia galanga*, *Zingiber purpureum*		
	Analgesic	Herbal baths of the rhizome	External	
	Hemorrhage	Rhizomes paste		[46]
	Aphrodisiac	Rhizomes	Oral	[47]
	Hallucination	Rhizomesare mixed with Indian hemp and *Podophyllum pleianthum*		[48]
	Fair skin	Leaves of *A. calamus* are given with *Artemisia vulgaris*	Dermal	[49]
Indonesia	Gastrointestinal	Rhizomes	Oral	[50]
England		Rhizomes blended with chalk and magnesium oxide		[51]
	Gastrointestinal, antibacterial, analgesic	Rhizomes		[52]
	Neurological, dysentery, and chronic catarrh	Rhizomesare given with *Gentiana campestris* L.		
	Malaria	Rhizomes		[53]
Europe	Obesity, influenza, gastrointestinal, respiratory			[54,55]
Republic of South Africa	Tooth powder, gastrointestinal, tonic, aphrodisiac			[56]
Sweden	Liquor			[57]
Germany	Increases menstrual flow, gastrointestinal			[58,59]
Java	Lactation			[60]
Lithuania	Chest pain, diarrhea	Rhizomes and leaves are taken with sugar		[52]
	Relieves pain, gout, rheumatism	Leaves decoction	External	[61]
New Guinea	Miscarriage	Rhizomes	Oral	[62]
Philippines	Gastrointestinal, rheumatism			[56]
Russia	Typhoid, syphilis, baldness, fever, cholera			[63]
Thailand	Blood purifier, fever			[64]
Turkey	Wound healing, cough, tuberculosis		External and oral	[61]
	Gastrointestinal		Oral	[65,66]
Arab countries	Gastrointestinal, tuberculosis			[67,68]
Brazil	Destroys parasitic worms			[68]
Argentina	Dysmenorrhea			[69]
United States	Gastrointestinal, abortifacient, stimulant, tonic, respiratory disorder	Rhizomes		[70]
Korea	Improves memory and life span			[71]
Sri Lanka	Cough, worm infestation	Rhizomes paste are given with milk		[72]

**Table 2 jcm-09-01176-t002:** Pharmaceutical products of *A. calamus* available in the market.

Medicine/Formulations	Indications/Use	Manufacturers
Pilochek tablets	Hemorrhoids	Dabur India Limited
Brahm Rasayan	Nervine tonic	
Mahasudarsan Churna	Malaria	
Janma Ghunti Honey	Babies growth, Constipation, Diarrhea	
Brahmi Pearls capsules	Brain Nourisher	Kerala Ayurveda
GT capsules	Osteoarthritis, osteoporosis, hyperlipidemia	
Histantin tablets	Anti-allergic	
Santhwanam oil	Antioxidant, rejuvenate	
Mahathikthaka Ghrita capsules	Skin disease, malabsorption syndrome	
Calamus root tincture	Stimulates the digestive system	Florida Herbal Pharmacy
Vacha capsules	Food supplements	DR Wakde’s Natural Health Care, London
Mentat tablets and syrup	Nervine tonic	Himalaya Herbal Healthcare
Abana	Cardiovascular disorders, hyperlipidemia, dyslipidemia	
Mentat tablets and Syrup	Anxiety, depression, insomnia	
Muscle & Joint Rub	Backaches, muscular sprains, pain	
Anxocare	Anxiety	
Erina-EP	Ectoparasites	
Himpyrin, Himpyrin Vet	Analgesic and anti-inflammatory	
Scavon Vet	Anti-bacterial, anti-fungal	
Vacha powder	Brain tonic, improves digestion, and prevents nausea	Bixa Botanical
Amalth	Herbal supplements	Mcnow Biocare Private Limited
Sunarin capsules	Anal fissures, piles, rectal inflammation, congestion	SG Phyto Pharma
Dr Willmar Schwabe India *Acorus calamus* mother tincture	Intestinal worms and stomach disorders, fever, nausea	Dr Willmar Schwabe India Pvt Ltd.
Himalayan calamus root essential oil	Pain relief and calm mind	Naturalis Essence of Nature
Calamus oil	Body, skin care, hair growth	Kazima Perfumers
Calamus root powder	Mental health problems	Heilen Biopharm
Winton tablets and syrup	Reduce tension, stress, and anxiety	Scortis Healthcare
Chesol syrup	Muscular aches and pains, chest colds, and bronchitis	J & J Dechane Laboratories Private Limited
Enzo Fast	Acidity, gastritis, flatulence, indigestion	Naturava
Dark Forest Vekhand powder	Abdomen pain, worms (infants)	Simandhar Herbal Pvt. Ltd.
Nervocare	Insomnia	Deep Ayurveda
Antress tablets	Anxiety and stress disorders	Ayursun Pharma
Grapzone syrup	Mental wellness	Alna Biotech Pvt Ltd.
Memoctive syrup	Improves memory power	Aayursh Herbal India
Smrutihills capsules	Stress, anxiety, adaptogenic	Ayush Arogyam
Gastrin capsules	Gastritis, dyspepsia	Sarvana Marundhagam
Pigmento tablets	Leukoderma or vitiligo	Charak Pharma
Paedritone drops	Digestive functions	
Vacha Churna	Brain tonic, digestion, nausea	Sadvaidyasala
Alert capsules	Immunomodulator, anxiety	Vasu Healthcare
Brento tablets	Increasing cognitive functions	Zandu Realty Limited
Livotrit Forte	Hepatitis, jaundice	
Zanduzyme	Indigestion and dyspepsia	
Vedic Slim	Anti-obesity	Vedic Bio-Labs Pvt. Ltd.
Hinguvachaadi Gulika	Anorexia, indigestion, appetite loss	Nagarajuna Pvt. Ltd.
Nilsin capsules	Sinusitis and allergic rhinitis	Phytomarketing
Norbeepee tablet	Hypertension	AVN Formulations
Sooktyn tablet	Antacid, antispasmodic	Alarsin Pharma Pvt. Ltd.
Deonac oil	Pain reliving oil	Doux Healthcare Pvt. Ltd.
Smrutisagar Rasa	Memory enhancer	Shree Dhootpapeshwar Limited
Yogaraj Guggul	Vitiligo, anorexia, indigestion, loss of appetite	
Kankayan Bati	Gastritis, flatulence, dyspepsia	Baidyanath Pvt. Ltd.
Brahmi Ghrita	Insanity and memory issues	
Fat Go	Controls high cholesterol level	Jolly Healthcare
Divya Medha Vati	Improves memory power	Patanjali Ayurveda
Divya Mukta Vati	High blood pressure	

**Table 3 jcm-09-01176-t003:** Chemical compounds isolated from different botanical parts of *A. calamus*.

Classification	Compound No.	Chemical Ingredient	Methods of Characterization	Parts/Extract	References
Phenylpropanoids	**1 **	*α*-Asarone	GC-FID, GC-MS	Rhizomes/n-hexane, aqueous, methanol, ethanol	[74,78,84,89,90,91]
	**2 **	*β*-Asarone			
	**3 **	*γ*-Asarone			
	**4 **	Eugenyl acetate	GC-MS	Rhizomes/aqueous extract	[74,78,91]
	**5 **	Eugenol			
	**6 **	Isoeugenol			
	**7 **	Methyl eugenol		Rhizomes/n-hexane, ethyl acetate	[92]
	**8 **	Methyl isoeugenol		Rhizomes/hexane	[74,78,91,94]
	**9 **	Calamol		Rhizomes/aqueous extract	[74,78,91]
	**10 **	Azulene			
	**11 **	Eugenol methyl ether			
	**12 **	Dipentene			
	**13 **	Asaronaldehyde			
	**14 **	Terpinolene			
	**15 **	1,8-cineole			
	**16 **	(*E*)-isoeugenol acetate	GC-FID, GC-MS		[89]
	**17 **	(*E*)-methyl isoeugenol			
	**18 **	Cis-methyl isoeugenol		Rhizomes/n-hexane, ethyl acetate	[92]
	**19 **	Euasarone			
	**20 **	Cinnamaldehyde			
	**21 **	Cyclohexanone	GC-MS	Rhizomes/hexane	[94]
	**22 **	Acorin	NMR	Rhizomes/chloroform	[95]
	**23 **	Isoasarone			
	**24 **	Safrole			
	**25 **	Z-3-(2,4,5-trimethoxyphenyl)-2-propenal	FTIR, NMR	Rhizomes/ethanol	[96]
	**26 **	2,3-dihydro-4,5,7-trimethoxy-1-ethyl-2-methyl-3 (2,4,5-trimethoxyphenyl) indene			
	**27 **	(Z)-asarone	GC-MS	Leaves/n-hexane	[97]
	**28 **	(E)-caryophyllene			
	**29 **	Estragole		Rhizomes/aqueous	[98]
	**30 **	Carvacrol			
	**31 **	2-cyclohexane-1-one			
	**32 **	Naphthalene			
	**33 **	*γ*-Cadinene			
	**34 **	Aristolene			
	**35 **	1(5),3-aromadenedradiene			
	**36 **	5-n-butyltetraline			
	**37 **	4,5-dehydro-isolongifolene			
	**38 **	Calarene			
	**39 **	Isohomogenol			
	**40 **	Zingiberene			
	**41 **	*α*-Calacorene			
	**42 **	5,8-dimethyl isoquinoline			
	**43 **	Cyclohexane methanol			
	**44 **	Longifolene			
	**45 **	Isoelemicin			
Sesquiterpenoids	**46 **	Calamene			[74,78,91]
	**47 **	Calamenenol			
	**48 **	Calameone			
	**49 **	Preisocalamendiol			
	**50 **	1,4-(trans)1,7(trans)-acorenone			[93]
	**51 **	1,4-(cis)-1,7-(trans)-acorenone			
	**52 **	2,6 diepishyobunone			
	**53 **	*α*-Gurjunene			
	**54 **	*β*-Gurjunene			
	**55 **	*α*-Cedrene			[98]
	**56 **	*β*-Elemene			
	**57 **	*β*-Cedrene			[93]
	**58 **	*β*-Caryophyllene			
	**59 **	Valencene			
	**60 **	Viridiflorene			
	**61 **	*α*-Selinene	GC-FID, GC-MS		[89,93]
	**62 **	δ-Cadinene	GC-MS		[93]
	**63 **	α-Curcumene			
	**64 **	Shyobunone			[84,93,99,100]
	**65 **	Isoshyobunone			[93,99,101]
	**66 **	Caryophyllene oxide			[93]
	**67 **	Humulene oxide II	GC-FID, GC-MS		[89,93]
	**68 **	Elemol	GC-MS		[93]
	**69 **	Cedrol			
	**70 **	Spathulenol			
	**71 **	Acorenone			
	**72 **	*α*-Cadinol			
	**73 **	Humulene epoxide II	GC-FID, GC-MS		[89]
	**74 **	*α*-Bisabolol			
	**75 **	Asaronaldehyde	NMR	Rhizomes/chloroform	[95]
	**76 **	Calamusenone	GLC, IR, NMR	Rhizomes/petroleum ether	[99]
	**77 **	Isocalamendiol			
	**78 **	Dehydroxyiso-calamendiol			
	**79 **	Epishyobunone			
	**80 **	Acorone	NMR	Rhizomes/hydro alcoholic	[100]
	**81 **	Neo-acorane A		Rhizomes/ethanol	[102]
	**82 **	Acoric acid			
	**83 **	Calamusin D			
	**84 **	1*β*,5*α*-Guaiane-4*β*,10*α*-diol-6-one			[103]
	**85 **	Dioxosarcoguaiacol	HPLC	Rhizomes/petroleum ether	[101]
	**86 **	7-tetracycloundecanol,4,4,11,11-tetramethyl	GC-MS	Rhizomes/ethanol	[84]
	**87 **	4*α*,7-Methano-4α-naphth[1,8a-b] oxirene,			
	**88 **	Spathulenol		Rhizomes/aqueous	[98]
	**89 **	Vulgarol B			
	**90 **	Tatanan A	HPLC, NMR	Rhizomes/95% ethanol	[104]
	**91 **	Acoramone			
	**92 **	2-hydroxyacorenone			
	**93 **	4-(2-formyl-5-methoxymethyl pyrrol-1-yl) butyric acid methyl ester			
	**94 **	2-acetoxyacorenone			
	**95 **	Acoramol			
	**96 **	N-transferuloyl tyramine			
	**97 **	Tatarinoid A			
	**98 **	Tatarinoid B			
	**99 **	Acortatarin A			
Monoterpenes	**100 **	*α*-Pinene	GC-MS	Rhizomes, roots/aqueous	[74,78,91,93]
	**101 **	*β*-Pinene			
	**102 **	Camphene			[74,78,91,93,98]
	**103 **	o-Cymol			[98]
	**104 **	p-Cymene	GC-FID, GC-MS		[89,93,98]
	**105 **	*γ*-Terpinene	GC-MS		[98]
	**106 **	*α*-Terpinolene			
	**107 **	Anethole			
	**108 **	Thymol			
	**109 **	Isoaromadendrene epoxide			
	**110 **	Camphor		Rhizome, leaves, roots/aqueous, hexane	[93,97]
	**111 **	Sabinene		Roots/aqueous	[93]
	**112 **	2-hexenal			
	**113 **	Limonene			[93,98]
	**114 **	Cis-linaloloxide			[93]
	**115 **	Cis-sabinene hydrate			
	**116 **	Trans-linalol oxide			
	**117 **	Linalool			[93,97]
	**118 **	Terpinen-4-ol			[93]
	**119 **	*α*-Acoradiene			
	**120 **	*β*-Acoradiene			
	**121 **	*α*-Terpineol			
	**122 **	Isoborneol		Leaves/hexane	[97]
Xanthone glycosides	**123 **	4,5,8-trimethoxy-xanthone-2-*O*-*β*-D-glucopyranosyl (1-2)-*O*-*β*-D-galactopyranoside	NMR	Rhizome/ethanol	[83]
Triterpenoid saponins	**124 **	1*β*,2*α*,3*β*, 19*α*-Tetrahydroxyurs-12-en-28-oic acid-28-O- {(*β*-D-glucopyranosyl (1-2)}-*β*-D galactopyranoside			[82]
	**125 **	3-*β*, 22-*α*-24,29-Tetrahydroxyolean-12-en-3-O-(*β*-Darabinosyl (1,3)}-*β*-D-arabinopyranoside			
Alkaloids	**126 **	Trimethoxyamphetamine,2,3,5	GC-MS		[84]
	**127 **	Pyrimidin-2-one,4-[N-methylureido]-1-[4methyl amino carbonloxy methy]			
Triterpene glycoside	**128 **	22-[(6-deoxy-*α*-L-rhamnopyranosyl) oxy]-3,23-dihydroxy-, methyl ester, (3*β*,4*β*,20*α*,22*β*)	NMR	Root, Rhizomes/ethyl acetate	[85]
Steroids/Sterols	**129 **	*β*-daucosterol			
Amino acids	**130 **	Arginine	HPLC	Roots/ethanol	[86,87]
	**131 **	Lysine			
	**132 **	Phenylalanine			
	**133 **	Threonine			
	**134 **	Tryptophan			
	**135 **	*α*-alanine			
	**136 **	Asparagine			
	**137 **	Aspartic acid			
	**138 **	Norvaline			
	**139 **	Proline			
	**140 **	Tyrosine			
	**141 **	Glutamic acid			
Fatty acids	**142 **	Palmitic acid	GLC	Rhizome/petroleum ether	[88]
	**143 **	Myristic acid			
	**144 **	Palmitoleic acid			
	**145 **	Stearic acid			

GC-FID, gas chromatography – flame ionization detector; GC-MS, gas chromatography – mass spectrometry; NMR, nuclear magnetic resonance; FTIR, Fourier-transform infrared spectroscopy; GLC, gas liquid chromatography; IR, infrared spectroscopy; HPLC, high-performance liquid chromatography.

**Table 4 jcm-09-01176-t004:** Preclinical claims of *A. calamus* in neurological and metabolic disorders.

Action	Parts of Plant	Extract/Compound	Animal Model	Dosage	Results	References
Antidiabetic effects	Rhizomes	Methanol	STZ-induced	50, 100, and 200 mg/kg, p.o. to rats	↓ Lipid profile and blood glucose, while ↑ levels of plasma insulin, tissue glycogen, and G6PD	[105]
			Alloxan-induced	150 and 200 mg/kg, p.o. to rat	↓ Blood glucose level	[106]
		Ethyl acetate	Genetically obese diabetic C57BL/Ks db/db mice	100 mg/kg, p.o.	↓ Levels of triglycerides and serum glucose	[107]
			GLP-1 expression and secretion with STZ-induced	100 mg/kg, i.g.	↑ Secretion of GLP-1 and ↓ blood glucose levels	[108]
			In vitro HIT-T15 cell line and alpha-glucosidase enzyme	6.25, 12.5, and 25 µg/mL	↑ Insulin secretion in HIT-T15 cells	[109]
			Glucose tolerance	400 and 800 mg/kg, p.o. to mice	↓ Serum glucose, and abolished the ↑ level of blood glucose	
Anti-obesity effects		Ethanol and aqueous	HFD-induced	100 and 200 mg/kg to rats	↓ Levels of serum cholesterol and triglycerides, ↑ lipoprotein fraction	[110]
		Diethyl ether	HFD-induced	20 and 40 mg/kg, p.o. to rats	↓ Total cholesterol and low-density lipoprotein levels, ↑ plasma fibrinogen levels	[111]
		Methanol	Triton-X-100-induced hyperlipidemic	250 and 500 mg/kg to rats	Dose-dependent anti-hyperlipidemic effect	[112]
			HFD-induced	250 and 500 mg/kg, p.o. to rats	↓ Level of total cholesterol, triglycerides, and LDL, ↑ HDL cholesterol	[113]
		Aqueous	HFD-induced	100, 200, and 300 mg/kg, p.o. to rats	↓ Levels of serum glucose, leptin, and insulin along with ↓ triglyceride, low-density lipoprotein, very LDL cholesterol, total cholesterol, phospholipids, and free fatty acid increased levels	[114]
Antihypertensive effects		Ethyl acetate	Clamping the left kidney artery for 4 h	250 mg/kg, p.o. to rats	↓SBP and DBP, blood urea nitrogen, creatinine and LPO, ↑ level of nitric oxide, SOD, CAT, GPX	[115]
		Crude extract, ethyl acetate and n-hexane	Blood pressure lowering effect in normotensive	10, 30, and 50 mg/kg to anesthetized rats	Relaxant effects mediated through Ca^+2^ antagonism and NO pathways	[116]
		Ethanol and α-asarone	Dimethyl sulfoxide-induced noise stress to rats	100 and 9 mg/kg, p.o. to rats	↓ Destructive effect of stress enlightening the morphological changes of hippocampus	[117]
Anti-inflammatory effects	Leaves	Ethanol	Carrageenan-induced paw edema	100 and 200 mg/kg to rats	↓ Histamine, 5-HT, and kinins	[118]
Antioxidant effects	Rhizomes	*α*-asarone	Noise stress induced to rats	3, 6, and 9 mg/kg, i.p. to rats	↑ SOD and LPO, decreased ↓ CAT, GPX, GSH, vitamins C and E, and protein thiol levels	[119]
	Leaves and rhizomes	Ethyl acetate and methanol	DPPH radical scavenging chelating ferrous ions, FRAP	200, 100, 80, 60, 40, 20, 10, and 5 μg/mL	Prominent DPPH scavenging activity, chelating ferrous ions, and reducingpower	[120,121]
	Rhizomes	Ethanol	Acetaminophen-induced	250, 500 mg/kg, p.o. to rats	↓ MDA and ↑ SOD, CAT, GPX, GSH levels	[122]
Anticonvulsant effects	Roots	Ethanol and *β*-asarone	Kainic acid-induced convulsion	35 and 20 mg/kg	↓ Epileptic seizure, neuroprotective, and regenerative ability	[123]
		Methanol	PTZ-induced convulsion	100 and 200 mg/kg, p.o. to mice	↑ Latency period and ↓ PTZ-induced seizure time	[124]
	Rhizomes	Calamus oil	MES, PTZ, and MCS model	30, 100, and 300 mg/kg, p.o. to mice	Calamus oil is found stable	[125]
		Ethanol	MES and PTZ-induced convulsion	250, 500 mg/kg, p.o. to mice	↓ Hind limb extension and tonic flexion of forelimbs	[126]
		Methanol	MES and PTZ-induced	250 and 150 mg/kg, p.o. to rats	↓ Immobility time at 250 mg/kg; however, ineffective at 150 mg/kg	[127]
Antidepressant effects			TST and FST	50 and 100 mg/kg, i.p. to mice	↓ Immobility time in a dose-dependent manner	[128]
	Leaves		TST and FST	50 and 100 mg/kg	↓ Immobility time	[129]
	Roots	Aqueous	TST and FST	100, 150, 200 mg/kg, p.o. to mice	↓ Immobility time	[130]
	Rhizomes	Hydro-alcoholic extract	TST and FST	75 and 150 mg/kg, p.o. to mice	↓ Corticosteroid levels	[131]
		Ethanol	OFB and HPM test	72 mg/kg, p.o.	No stimulation of postsynaptic 5-HT1A receptors	[132]
		Methanol and acetone	Behavioral despair test	5, 20, and 50 mg/kg, p.o.	↓ Spontaneous locomotor activity	[133]
		*β*-asarone	EPM and FST	25, 50, and 100 mg/kg, p.o.	↓ Immobility time	[134]
Neuroprotective effects		Hydro-alcoholic	CCI of sciatic nerve-induced neuropathic pain	10 mg/kg to rats	Significantly ameliorated CCI-induced nociceptive pain	[135]
			CCI of sciatic nerve-induced peripheral neuropathy	100 and 200 mg/kg to rats	Prevented CCI-induced neuropathy through ↓ oxidation and inflammation	[136]
	Leaves	Methanol and acetone	Apomorphine-induced stereotypy and haloperidol-induced catalepsy	20 and 50 mg/kg to mice	Reversed stereotypy induced by apomorphine and significantly potentiated catalepsy induced by haloperidol	[137]
	Rhizomes	Ethanol	Spontaneous electrical activity and monoamine levels of the brain	200 and 300 mg/ kg to rats	Depressive response by altering electrical activity, including changing brain monoamine levels	[138]
		Hydro-alcoholic	MCAo-produced brain ischemia	25 mg/kg to rats	Improvement in neurobehavioral performance, ↓ levels of GSH, SOD, and ↑ LPO level	[139]
		Ethanol	Methotrexate-induced stress	5, 10, 15, 20, 25 ppm concentration to fruit flies	↓ Elevated ROS, SOD, CAT, and GPX levels	[140]
Cardioprotective effects	Whole plant		DOX-induced myocardial toxicity	100 and 200 mg/kg to rats	↓ Serum enzyme levels and protected the myocardium from the toxic effect of DOX	[141]
	Rhizomes	Crude, n-hexane, ethyl acetate	Guinea pig tracheal segments	0.01 mg/mL	↓ Force and rate of contractions at higher concentrations	[142]

CAT, catalase; CCI, chronic constriction injury; COX, cyclooxygenase; DBP, diastolic blood pressure; DOX, doxorubicin; DPPH, 2,2-diphenyl-1-picrylhydrazyl radical; EPM, elevated plus maze; FRAP, ferric reducing antioxidant power; FST, forced swim test; GLP-1, glucagon-like peptide-1; GPX, glutathione peroxidase; GR, glutathione reductase; GSH, reduced glutathione; HDL, high-density lipoproteins; HFD, high-fat diet; HPM, high plus maze; i.g., intragastric; i.p., intraperitoneal; LDL, low-density lipoprotein; LPO, lipid peroxides; MCAo, middle cerebral artery occlusion; MCS, minimal clonic seizure; MDA, malondialdehyde; MES, maximal electroshock; NO, nitric oxide; OFB, open field behavior; p.o., per oral; PTZ, pentylenetetrazol; ROS, reactive oxygen species; SBP, systolic blood pressure; SOD, superoxide dismutase; STZ, streptozotocin; TST, tail suspension test.

**Table 5 jcm-09-01176-t005:** Clinical claims of *A. calamus* in neurological and metabolic disorders.

Formulations/Dosage forms *A. calamus*	Subjects	Study Design	Intervention	Primary Endpoint	Outcome	Evidence Quality	Reference
*A. calamus* rhizome powder	24 patients of both sexes with hyperlipidemia	Randomized single-blind controlled study	500 mg twice daily after meal for 1 month	BMI, body perimeter, skinfold depth	Significant reduction in skinfold depth, fatigue, and excessive hunger	III	[178]
Davaie Loban capsules (*A. calamus*, nut grass, incense, ginger, and black pepper)	24 patients of both sexes with Alzheimer’s disease	Double-blind randomized clinical study	500 mg capsule thrice daily for 3 months	ADAS-cog and CDR-SOB scores	At 4 weeks and 12 weeks: significant reduction in the ADAS-cog and CDR-SOB scores	III	[179]
70% hydro-alcoholic extract of *A. calamus*	33 patients of both sexes (20 male and 13 female) with anxiety disorder	Non-randomized, open-label, single-arm study	500 mg extract of one capsule twice daily after meal for 2 months	BPRS score	Significant reduction of anxiety and stress-related disorder	III	[180]
Vachadi Churna (*A. calamus*, *Cyperus rotundus*, *Cedrus deodara*, ginger, *Aconitum Heterophyllum*, *T. chebula*)	30 obese patients of both sexes aged 14–50 years	Non-randomized, open-label, single-arm study	3 g powder twice daily with lukewarm water before meal for 1 month	BMI, girth measurements of mid-thigh, abdomen, hip, chest	Significant improvement in extreme sleep, body heaviness, fatigue, and excessive hunger	III	[181]
Guduchyadi Medhya Rasayana, (*A. calamus*, *Tinospora cordifolia*, *Achyranthes aspera*, *Embelia ribes*, *Convolvulus pluricaulis*, *T. chebula*, *S. lappa*, *Asparagus racemosus*, cow ghee, and sugar)	138 patients of both sexes aged 55–75 years with senile memory impairment	Randomized, two-parallel-group study	3 g granule thrice daily after meal for 3 months	Mini–Mental State Examination, BPRS score, and estimation of serum acetylcholinesterase	Significant improvement in terms of recall memory, cognitive impairment, amnesia, concentration ability, depression, and stress	III	[182]
Dried aqueous extract of *A. calamus*	40 healthy volunteers, both sexes aged 18–50 years with a premedicant for anesthesia	Open-label randomized, two- parallel-group study	90 min before anesthesia; In the control group: 0.2 mg intramuscular (IM) glycopyrrolate and a 0.2 mg IM 50 mg tablet of promethazine hydrochloride with water; In the second group: 0.2 mg IM glycopyrrolate and 100 mg *A. calamus* extract	Pulse rate, blood pressure, respiratory rate, body temperature	The dried aqueous extract exhibited anti-hyperthermic and sedative effect without producing any respiratory depression	III	[183]
Shankhapushpyadi Ghana Vati (*A. calamus*, *C. pluricaulis*, *Bacopa monnieri*, *T. cordifolia*, *C. fistula*, *A. indica*, *S. lappa*, *Tribulus terrestris*)	20 hypertensive patients of both sexes	Randomized single-blind controlled study	1 g twice daily after meal for 2 months	SBP and DBP	Significant relief in raised SBP and DBP	III	[184]
Brahmyadiyoga (*A. calamus*, *Centella asiatica*, *Rauvolfia serpentina*, *Saussurea lappa*, *Nardostachys jatamansi*)	10 schizophrenia patients of both sexes aged 18–40 years	Non-randomized, open-label, single- arm study	4 tablets thrice daily for three months after meal	Symptoms rating scale	Significant effect as a brain tonic, tranquillizer, hypnotic, and sedative	III	[185]
Bala compound (*A. calamus*, *Emblica officinalis*, *E. ribes*, *T. cordifolia*, *Piper longum*, *Glycyrrhiza glabra*, *C. rotundus*, *A. heterophyllum*)	24 neonates, both sexes, 2.5–3 kg body weight	Randomized single-blind controlled study	5 oral drops twice daily for 6 months	Change in serum immunoglobulins (IgG, IgM, and IgA) levels	Significant improvement in immunoglobulin levels after 6 months	Ib	[186]
Vachadi Ghrita (*A. calamus*, *T. cordifolia*, *Hedychium spicatum*, *C. pluricaulis*, *E. ribes*, ginger, *A. aspera*, *T. chebula*, and cow ghee)	90 healthy individuals of both sexes aged 40–50 years for assessment of cognition	Non-randomized positive-controlled study	10 g twice daily for 1 month with lukewarm water	Post Graduate Institute Memory Scale (PGIMS) test	Significant change in the mental balance score, holding of like and different pairs, late-immediate memory, and also improved digestion	III	[187]
Bramhi Vati (*A. calamus*, *B. monnieri*, *C. pluricaulis*, *Onosma bracteatum*, copper pyrite, iron pyrite, mercuric sulphide, *Piper nigrum*, *N. jatamansi*)	68 essential hypertension patients of both sexes aged 20–70 years	Randomized, double-blind, parallel-group comparative study	500 mg tablets twice daily for 1 month	Hamilton anxiety rating scale, SBP and DBP, and MAP	Significant improvement in the Hamilton anxiety rating scale, SBP and DBP, and MAP	III	[188]
Tagaradi Yoga (*A. calamus*, *Valeriana wallichii*, *N. jatamansi*)	24 insomnia patients of both sexes aged 18–75 years	Non-randomized positive-controlled study	500 mg hydro-alcoholic extract capsule twice daily after meal for 15 days	Sleep duration, initiating time of sleep, quality of sleep	Significant improvement in sleep duration, in the initiating time of sleep, and in quality of sleep	III	[189]
*Acorus calamus* rhizome powder	20 obese patients of both sexes	Randomized single-blind study	250 mg rhizome powder twice daily for 1 month	Body weight, height according to age, waist-hip ratio, and BMI	Significant improvement in extreme sleep, body heaviness, fatigue, and excessive hunger	III	[190]
*Acorus calamus* rhizome powder	45 ischemic heart disease patients	Non-randomized positive-controlled study	3 gm rhizome powder twice daily for 3 months	ECG, serum cholesterol level	Improvement of chest pain, dyspnea on effort, reduction of the body mass index, improved ECG: reduced serum cholesterol, reduced serum LDL, and increased serum HDL	Ib	[191]

ADAS-cog, alzheimer’s disease assessment scale–cognitive subscale; BMI, body mass index; BPRS, brief psychiatric rating scale; CDR-SOB, clinical dementia rating scale sum of boxes; DBP, diastolic blood pressure; ECG, electrocardiogram; Ib, evidence from at least one randomized study with control; HDL, high-density lipoprotein; Ig, immunoglobulin; III, evidence from well-performed nonexperimental descriptive studies, as well as from comparative studies, correlation studies, and case studies; LDL, low-density lipoprotein; MAP, mean arterial pressure; SBP, systolic blood pressure.

**Table 6 jcm-09-01176-t006:** Mechanistic role of phytochemicals of *A. calamus* in the treatment of neurological and metabolic disorders.

Study	Compound	Model	Increased Level	Decreased Level	References
Anti-Parkinson	*β*-Asarone	6-OHDA parkinsonian	Bcl-2 expression	GRP78, p-PERK, CHOP, and Beclin-1 expression	[192]
		6-OHDA parkinsonian	-	mRNA levels of GRP78 and CHOP and p-IRE1and XBP1	[193]
		Dopamine in the striatum	TH plasma concentrations	Striatal COMT levels	[194]
		6-OHDA parkinsonian	L-DOPA, DA, DOPAC, and HVA levels	P-gp, ZO-1, occludin, actin, and claudin-5	[195]
Alzheimer’s		A*β*25-35-induced inflammation	Bcl-2 level	TNF-*α*, IL-1*β*, IL-6, Beclin-1, and LC3B level	[196]
		NG108 cells	-	Upregulated SYP and GluR1 expression	[197]
		PC12 cells	-	A*β*-induced JNK activation, Bcl-w and Bcl-xL levels, cytochrome c release, and caspase-3 activation	[198]
		A*β*-induced cytotoxicity	Cell viability, p-Akt and p-mTOR	NSE levels, Beclin-1 expression	[199]
Neuroprotective		Pb-induced impairments	NR2B protein expression along with Arc/Arg3.1 and Wnt7a mRNA levels	-	[200]
	*β*-Asarone, eugenol	Scopolamine-induced	Improvement of neuron organelles and synaptic structure	APP expression	[201]
	Neotatarine	MTT reduction assay	-	A*β*25-35–induced PC12 cell death	[202]
	*β*-asarone, paeonol	MCAo model	Cholecystokinin and NF-κB signaling	TNF-*α*, IL-1*β*, IL-6 production	[203]
	*β*-Asarone	Cultured rat astrocytes	NGF, BDNF, and GDNF expression	-	[204]
		SN4741 cells	p62, Bcl-2 expression	JNK, p-JNK and Beclin-1 expressions	[205]
	Tatarinolactone	hSERT-HEK293 cell line	-	SERTs activity	[206]
	*β*-Asarone	RSC96 Schwann cells	GDNF, BDNF, and CNTF expression	-	[207]
		A*β*-induced	p-mTOR and p62 expression	AChE and A*β*_42_ levels, p-Akt, Beclin-1, and LC3B expression, APP mRNA and Beclin-1 mRNA levels	[208]
		A*β*1–42-induced injury	-	GFAP, AQP_4_, IL-1*β*, and TNF-*α* expression	[209]
Anti-depression		Chronic unpredictable mild stress	BDNF expression	Blocked ERK1/2-CREB signaling	[210]
	*α*-Asarone	Noradrenergic and serotonergic neuromodulators in TST	*α*_1_ and *α*_2_ adrenoceptors and 5-HT_1A_ receptors	-	[211]
Anticonvulsant and sedative	Eudesmin	MES and PTZ	GABA contents, expressions of GAD65, GABAA, and Bcl-2	Glu contents and ratio of Glu/GABA, caspase-3	[212]
Anti-anxiety	*α*-Asarone	BLA or CFA-induced	Down-regulation of GABA_A_ receptors	Up-regulation of GluR1-containing AMPA, NMDA receptors	[213]
Anti-epilepsy		Temporal lobe epilepsy	Levels of GABA, GAD67, and GABAAR-mRNA expression	GABA-T	[214]
		Mitral cells	Down-regulation of GABA_A_ receptors	Na^+^ channel blockade	[215]
	*β*-Asarone	KA-induced	GABA	Glu	[216]
Anti-inflammatory	*α*-Asarone	Spinal cord injury	IL-4, IL-10, and arginase 1 levels	TNF-*α*, IL-1*β*, IL-6, MCP-1, MIP-2, iNOS levels	[217]
Cytoprotective	*β*-Asarone	tBHP-induced astrocyte injury	GST, GCLM, GCLC, NQO1, Akt phosphorylation	-	[218]
Cardioprotective		Cultured neonate rat cardiac myocytes	Viability of cardiac myocytes	Pulse frequency	[219]
Arteriosclerosis		ECV304 cell strain	Apoptotic rate of ECV304 cells	Apoptotic rate of MMP, stabilized MMP and VSMC proliferation	[220]
Anti-adipogenic		3T3-L1 preadipocytes	-	C/EBP*β*, C/EBP*α*, and PPAR*γ* expression levels, ERK1/2 phosphorylation	[89]
Antioxidant		Cerebral artery occlusion	Antioxidant activity	Focal cerebral ischemic/reperfusion injury	[221]
Anti-diabetic	*α*-Asarone + *β*-asarone + metformin HCl	STZ-induced	Insulin level	Glucose, glycosylated hemoglobin level, liver dysfunction, and tumor biomarkers	[222]
	Asarone	3T3-L1 preadipocytes	Hormone-sensitive lipase phosphorylation	Intracellular triglyceride levels, down-regulation of PPAR*γ* and C/EBP*α*	[223]

6-OHDA, 6-hydroxydopamine; Ox-LDL, oxidized low-density lipoprotein; BDNF, brain-derived neurotrophic factor; NGF, nerve growth factor; GDNF, glial derived neurotrophic factor; SERTs, serotonin transporters; MCAo, middle cerebral artery occlusion; A*β*, *β*-amyloid; NSE, neuron specific enolase; AMPA, *α*-amino-3-hydroxy-5-methyl-4-isoxazolepropionic acid; NMDA, NR2A-containing N-methyl-D-aspartate; GABA_A_, *γ*-aminobutyric acid A; BLA, basolateral amygdala; CFA, complete Freund’s adjuvant; CNTF, ciliary neurotrophic factor; COMT, catechol-O-methyltransferase; TH, tyrosine hydroxylase; DA, dopamine; DOPAC, 3,4-dihydroxyphenylacetic acid; HVA, homovanillic acid; P-gp, P-glycoprotein; ZO-1, zonula occludens-1; SYP, synaptophysin; GluR1, glutamatergic receptor 1; GABA-T, GABA transaminase; TST, tail suspension test; KA, kainic acid; MCP-1, monocyte chemoattractant protein 1; MIP-2, macrophage inflammatory protein 2; iNOS, inducible nitric oxide synthase; GST, glutathione S-transferase; GCLM, glutamate-cysteine ligase modulatory subunit; GCLC, glutamate-cysteine ligase catalytic subunit; NQO1, NAD(P)H quinone oxidoreductase; GFAP, glial fibrillary acidic protein; AQP, aquaporin; VSMC, vascular smooth muscle cells; MMP, mitochondrial membrane potential; C/EBP, CCAAT enhancer-binding protein; PPAR*γ*, peroxisome proliferator-activated receptor gamma; ERK1/2, extracellular signal-regulated protein kinase; XBP1, x-box binding protein; IRE1, inositol-requiring enzyme 1; Aβ1-42, amyloid β peptide; mTOR, mammalian target of rapamycin; MTT, 3-(4,5-dimethythiazol-. 2-yl)-2,5-diphenyl tetrazolium bromide; CREB, cAMP response element-binding protein; GABAAR, gamma-aminobutyric acid type-A receptor, tBHP, t-butyl hydroperoxide.

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
