# Peer review of "Role of Vacha (Acorus calamus Linn.) in Neurological and Metabolic Disorders: Evidence from Ethnopharmacology, Phytochemistry, Pharmacology and Clinical Study"

_jcm, 2020, doi:10.3390/jcm9041176_

Round 1
Reviewer 1 Report
The review article by Sharma et al provides a comprehensive overview on the medicinal roles of Acorus calamus in neurological and metabolic diseases. The review extensively covers the important aspects on this topic in well organized sections. My minor comments can be found below:
- When describing the antioxidant effect of A. calamus, the authors emphasize the effect it has on the Nrf2 pathway. Are there any effects on the cytochrome P450 signaling pathway as well since this is an important pathway that regulates anti-oxidative response in a cell? The authors might want to comment on this.
- While describing the anti-diabetic/cardioprotective effects, the authors mostly state the consequences and do not elaborate on the mechanism of action. How exactly is lipolysis affected, for example? Does it regulate important enzymes in the lipolytic cycle such as ATGL or PPARg?
- The authors could concise the section on the geographical distribution of A. calamus and focus more on the medicinal roles as suggested by the title
Author Response
The review article by Sharma et al provides a comprehensive overview on the medicinal roles of Acorus calamus in neurological and metabolic diseases. The review extensively covers the important aspects on this topic in well organized sections. My minor comments can be found below:
- When describing the antioxidant effect of A. calamus, the authors emphasize the effect it has on the Nrf2 pathway. Are there any effects on the cytochrome P450 signaling pathway as well since this is an important pathway that regulates anti-oxidative response in a cell? The authors might want to comment on this.
Response: Thanks for your valuable suggestion. We have now incorporated this point in the Manuscript (Added as point 6.10._ Cytochromes Inhibitory Activities).
- While describing the anti-diabetic/cardioprotective effects, the authors mostly state the consequences and do not elaborate on the mechanism of action. How exactly is lipolysis affected, for example? Does it regulate important enzymes in the lipolytic cycle such as ATGL or PPARg?
Response: Thank you for providing your useful and constructive suggestions. We have made a major revision which we hope meet requirement for publication (Added in point 8_Mechanistic role; Figure 7 is revised, Table 6 is added).
- The authors could concise the section on the geographical distribution of A. calamus and focus more on the medicinal roles as suggested by the title
Response: Thanks, the revised figure (Figure 4) is added in the manuscript as per your suggestion
Reviewer 2 Report
The Manuscript ID number “jcm-766887”, entitled "Role of Vacha (Acorus calamus Linn.) in neurological and metabolic disorders: Evidence from ethnopharmacology, phytochemistry, pharmacology and clinical study", is well written and fall with the scope of this Journal, I recommend the publication of the manuscript after minor revisions:
Line 38 change “hyperglycaemia” in hyperglycemia
Figure 6, revise the chemical structures and use the same font size for numbering
Line 187 and 188 “STZ-induced and 188 db/db diabetic mice” and “GLP-1 levels”, for clarity, write out the full term the first time you mention it, and put the abbreviation in parentheses after the name. please check acronyms through all the text
Line 265 and 268 please write A. calamus in italics
please check and revise all the bibliography: for example in Line 438 and 453 the journal is not in italics, while in line 476 and 477 the year is not in bold
Author Response
The Manuscript ID number “jcm-766887”, entitled "Role of Vacha (Acorus calamus Linn.) in neurological and metabolic disorders: Evidence from ethnopharmacology, phytochemistry, pharmacology and clinical study", is well written and fall with the scope of this Journal, I recommend the publication of the manuscript after minor revisions:
Line 38 change “hyperglycaemia” in hyperglycemia
Response: Thank you for your positive and constructive comments, we have revised the manuscript carefully to avoid such typo errors.
Figure 6, revise the chemical structures and use the same font size for numbering
Response: Thanks, we have rewritten it as per your suggestion.
Line 187 and 188 “STZ-induced and 188 db/db diabetic mice” and “GLP-1 levels”, for clarity, write out the full term the first time you mention it, and put the abbreviation in parentheses after the name. please check acronyms through all the text
Response: Thanks, we have rewritten it following your suggestion.
Line 265 and 268 please write A. calamus in italics
Response: Thanks, we have revised manuscript carefully to avoid such typo errors.
Please check and revise all the bibliography: for example in Line 438 and 453 the journal is not in italics, while in line 476 and 477 the year is not in bold
Response: Thanks, we have revised all the bibliography of the manuscript carefully to avoid such typo errors.